# Impact of Scala Tympani Geometry on Insertion Forces during Implantation

**DOI:** 10.3390/bios12110999

**Published:** 2022-11-10

**Authors:** Filip Hrncirik, Iwan V. Roberts, Chloe Swords, Peter J. Christopher, Akil Chhabu, Andrew H. Gee, Manohar L. Bance

**Affiliations:** 1Cambridge Hearing Group, Cambridge, UK; 2Department of Clinical Neurosciences, University of Cambridge, Cambridge CB2 0QQ, UK; 3Department of Physiology, Development and Neurosciences, University of Cambridge, Cambridge CB2 3DY, UK; 4Department of Engineering, University of Cambridge, Cambridge CB2 1PZ, UK; 5Clinical School, University of Cambridge, Cambridge CB2 0SP, UK

**Keywords:** cochlear implant, 3D printing, scala tympani, insertion forces, micro-CT

## Abstract

(1) Background: During a cochlear implant insertion, the mechanical trauma can cause residual hearing loss in up to half of implantations. The forces on the cochlea during the insertion can lead to this mechanical trauma but can be highly variable between subjects which is thought to be due to differing anatomy, namely of the scala tympani. This study presents a systematic investigation of the influence of different geometrical parameters of the scala tympani on the cochlear implant insertion force. The influence of these parameters on the insertion forces were determined by testing the forces within 3D-printed, optically transparent models of the scala tympani with geometric alterations. (2) Methods: Three-dimensional segmentations of the cochlea were characterised using a custom MATLAB script which parametrised the scala tympani model, procedurally altered the key shape parameters (e.g., the volume, vertical trajectory, curvature, and cross-sectional area), and generated 3D printable models that were printed using a digital light processing 3D printer. The printed models were then attached to a custom insertion setup that measured the insertion forces on the cochlear implant and the scala tympani model during a controlled robotic insertion. (3) Results: It was determined that the insertion force is largely unaffected by the overall size, curvature, vertical trajectory, and cross-sectional area once the forces were normalised to an angular insertion depth. A Capstan-based model of the CI insertion forces was developed and matched well to the data acquired. (4) Conclusion: By using accurate 3D-printed models of the scala tympani with geometrical alterations, it was possible to demonstrate the insensitivity of the insertion forces to the size and shape of the scala tympani, after controlling for the angular insertion depth. This supports the Capstan model of the cochlear implant insertion force which predicts an exponential growth of the frictional force with an angular insertion depth. This concludes that the angular insertion depth, rather than the length of the CI inserted, should be the major consideration when evaluating the insertion force and associated mechanical trauma caused by cochlear implant insertion.

## 1. Introduction

There are over 466 million people worldwide that suffer from disabling hearing loss and is expected to rise to 700 million by 2050 [1]. As the second leading disability worldwide [2], hearing loss can be severely debilitating and has been linked to depression [3,4,5,6,7], dementia [8,9,10,11], and living discomfort [12,13,14].

Those suffering from severe to profound sensorineural hearing loss can benefit from cochlear implants (CIs), a transformative technology that helps people regain their hearing. However, a key limitation in increasing the eligibility of CIs is the damage caused to the cochlea during the insertion of these implants as well as the resulting chronic inflammatory response. This insertion trauma has been shown to reduce or destroy the residual acoustic hearing in up to 50% of implantations [15,16,17,18,19]; therefore, only those with the most severe hearing loss are currently implanted.

A CI consists of a linear array of 12–22 platinum-based electrodes insulated by silicone connected via a lead to a processor placed on the skull. The electrode array is typically inserted into the *scala tympani* (ST) chamber of the cochlea, in the inner ear, which is a hollow spiral-shaped chamber within the mastoid bone, filled with perilymph fluid. In order to be implanted, patients must undergo surgery where the skin is opened behind the pinna, and the skull and mastoid bone are drilled until the facial nerve, chorda tympani, and incus are visible. These landmarks are used to find the round window, an opening to the ST covered with a membrane that can be easily penetrated and which provides an entry point for the CI insertion. The common approach is to implant through the round window niche and round window where possible, rather than through a cochleostomy [20,21,22,23,24,25], which requires surgeons to create a new entry to the ST by drilling into the cochlea. Implanted CIs can then electrically stimulate the auditory nerves in the modiolus (the inner part of the cochlea spiral) and facilitate “electronic hearing”.

Furthermore, by avoiding key auditory structures (i.e., the eardrum and the middle ear) during implantation, patients can retain some residual hearing and benefit from some limited auditory cues. Furthermore, electro-acoustic stimulation (EAS), which is a combination of CI electrically stimulated hearing and low-frequency acoustic amplification, has been shown to improve the listening performance [26,27,28,29]. Additionally, destroying the residual hearing eliminates the possibility of a patient potentially being eligible for future therapies to restore the acoustic hearing, such as gene therapies [30]. Therefore, it is necessary to protect patients’ residual hearing whenever possible.

The loss of natural hearing is often associated with the mechanical trauma that arises during the CI insertion [17,18,19,31,32], which might result in fibrosis and neural degeneration, further limiting the CI performance [18,33,34]. Generally, there are two types of CI: (1) straight, which follows the lateral wall of the ST, and (2) perimodiolar, which is pre-curved and follows the modiolar (inner) wall of the ST. Although perimodiolar electrodes can, in theory, be placed closer to the target neural population, their placement can also risk more insertion trauma [35]. For straight electrodes, the initial contact is with the lateral wall of the ST, which is lined with soft tissue, and the CI slides along the lateral wall throughout the insertion. The basilar membrane, located along the top part of the lateral wall, accommodates the organ of Corti, which facilitates acoustic hearing. It is crucial to avoid damaging, or penetrating in some circumstances, this membrane as it can result in the permanent loss of hearing in that region [36,37].

Although CIs are a triumphant story of permanently implantable devices, there are still limitations in the implantation process. For instance, CIs are commonly inserted manually by a skilled surgeon; however, it has been shown that the stable, slow insertion speeds achieved with a robotic and a semi-robotic insertion setup could lower insertion forces (IFs) significantly [33,38,39,40]. Furthermore, there has been uncertainty about the effect of the CI size and its influence on the IFs. Historically, there has been a trade-off between making longer implants that could electrically stimulate a larger proportion of the cochlea, and convey a larger range of frequencies [41,42], and preventing large IFs at deeper insertions. The higher IF and related mechanical trauma have led the CI industry to converge to a “one size fits all” approach where newer CI electrode arrays are typically around 20 mm in length [43,44]. Additionally, few studies investigated how the ST size may affect IFs [45,46,47,48]; however, these either used artificial models with combined scalae (not a true ST model) or they did not investigate the individual parameters that might contribute to higher IFs.

This study aims to investigate the impact of the ST geometry on IFs. By systematically adjusting the key parameters of the ST, such as the volume, vertical trajectory, curvature, and cross-section area, it is possible to determine the influence of these individual components on the IFs. Furthermore, this can inform the optimal CI insertion strategies to improve patient outcomes by creating a mechanistic model from the acquired insights that match our experimental observations.

## 2. Materials and Methods

### 2.1. Micro-CT Segmentation of Scala Tympani

A cadaveric specimen was imaged with a Nikon XT 225 ST micro-computed tomography (micro-CT) scanner with an accelerating voltage of 160 kV, current of 180 µA, and a voxel resolution of 27 µm. The specimen was reconstructed using Stradview software (version 7.0, https://mi.eng.cam.ac.uk/Main/StradView, accessed on 10 March 2022). Important landmarks highlighting 68 points along the basilar membrane on the ST surface were added, so complete parametrisation using a custom-built MATLAB script was possible.

### 2.2. Characterisation of Scala Tympani

The 3D PLY files exported from the segmentation were imported into a custom MATLAB script for analysis together with the coordinates of landmarks along the cochlear trajectory. This workflow used the gptoolbox [49] and geom3D [50] libraries available on the MathWorks FileExchange. Using a similar methodology to Gee et al. [51], a best-fit plane was determined for the landmark point coordinates along the first 270 degrees of the basal turn of the basilar membrane. This enabled the separation of the vertical (height) and radial (spiral) components of the cochlear trajectory.

The cross-sections of the ST model were determined by producing planes along the cochlear trajectory perpendicular to the cochlear lumen according to the landmarks placed along the ST and calculating the closest intersection of the plane and cochlear mesh to that landmark (to eliminate intersections through multiple turns of the cochlea).

This spiral, represented by the centroids of the cross-sections, was fitted to the following equation defining the radius of the curve, *R*, in terms of angle, *θ*, in degrees:(1)$$R\left(\theta \right)=R_{scale}\cdot \left(e^{\frac{-\theta }{\theta_{1}}}+e^{\frac{-\theta }{\theta_{2}}}\right)$$
where *R_scale_* parametrises the overall scale of the cochlea, and $\theta_{1}$ and $\theta_{2}$ characterise the tightness of the basal turn and central spiral, respectively. This is a modification of previous piecewise definitions [52,53,54] of the cochlear spiral in favour of a continuous double exponential function [55], which fits the whole cochlea while accounting for the “flaring out” of the base. Using a continuous function offers several advantages in the mathematical modelling and shape manipulation when compared to a piecewise function. Note, for the conversion of electrode insertion distance to degrees, Equation (1) was applied to the basilar membrane landmarks for each ST model as the CI would follow the lateral wall rather than the middle of the ST which was confirmed/adjusted according to the actual maximal insertion angle manually measured from the video of each insertion.

### 2.3. Manipulation of Scala Tympani Shape

In order to manipulate the shape of the cochlea, the extracted cross-sections described in Section 2.2 were positioned according to the required shape manipulation, as detailed below.

A custom lofting function was created in MATLAB to re-connect the cross-sections into a 3D mesh geometry from individual cross-sections. This involved sorting the vertices of the cross-sections in a clockwise direction (with respect to the base), interpolating the cross-sections to have 100 points each, triangulating the vertices between each cross-section in turn, and capping the ends to produce a fully enclosed mesh.

#### 2.3.1. Cochlear Size Manipulation

Volumetric scaling of the ST was conducted by directly multiplying the vertices of the original ST mesh by a constant factor to produce “large” (110% volumetric scaling) and “small” (90% volumetric scaling) models.

#### 2.3.2. Vertical Trajectory Manipulation

For vertical trajectory manipulation experiments, the z-component of the ST cross-sections was altered without changing the radial trajectory of the ST. For the “flat” models, the z-component of each cross-section was subtracted from each vertex so that the ST centreline would be a two-dimensional spiral along the basal plane.

In order to simulate non-planarity, a sinusoidal function was added to the mean z-component of each cross-section from 0–270°. The amplitude of the sinusoidal change was set at 200 µm, and the period was set at either 270° or 135° for conditions of artificial non-planarity 1 (NP1) and artificial non-planarity 2 (NP2), respectively.

#### 2.3.3. Curvature Manipulation

In order to manipulate the curvature of the ST models, the parameters of the fitted spiral Equation (1) were manipulated. Specifically, θ_2_ was doubled for the loose model and halved for the tight one to either decrease or increase the curvature of the inner spiral, respectively. The ST cross-sections were projected along this new spiral and lofted into a 3D structure ready to produce a 3D print.

#### 2.3.4. Uniform Cross-Section Models

Uniform cross-section models used a consistent cross-section (from 1 mm from the round window) projected onto the original ST centreline to build models with the same trajectory as the original ST but with a uniform cross-section.

### 2.4. 3D Printing an Artificial Scala Tympani

A custom MATLAB script, utilising Boolean operations from the gptoolbox library, was used to generate 3D printable stereolithography (STL) files for 3D printing. Scala tympani orientation was controlled to be consistent for all models where the first 10–20° degrees of the ST were orientated along the *x*-axis of insertion. In addition, the basal end of the ST remained open to provide a consistent entry trajectory, and an access hole was produced at the ST apex to allow for flushing of the models with solutions prior to insertion.

Prepared models were then printed at 30 µm resolution on a CADworks3D printer (M-50, CADworks3D μmicrofluidics, Toronto, ON, Canada) with Clear Microfluidics Resin (V7.0a, CADworks3D μmicrofluidics, Toronto, ON, Canada). The printed models were then post-processed using 99.9% isopropyl alcohol (SLS Ltd., Nottingham, UK). Lastly, the models were cured three times for 10 s with a one-minute break between the runs using a CureZone UV chamber (CADworks3D μmicrofluidics, Toronto, ON, Canada).

In order to achieve the transparent finish of the models, acrylic coating (Pro-cote Clear Laquer, Aerosol Solution) was used for coating the lumen (inner part) of the models. The coating was injected into the ST model, left for 10 s, and excess was then removed using compressed air to leave a thin layer to smooth the surface and achieve a clear finish. In addition, 5% solution of Pluronic (F-127, Merck KGaA, Germany) and distilled water was used for coating the lumen of the models 24 h prior to insertion to lower friction coefficient of the printed models.

### 2.5. Insertion Setup

A custom-built insertion setup used in this study consisted of several components, namely a one-axis force sensor (500 mN Load Cell, 402B, Aurora Scientific Europe), a si*x*-axis force sensor (NANO 17Ti transducer, ATI Industrial Automation, Apex, NC, USA), a one-axis motorised translation stage (PT1/M-Z8, Thorlabs, UK) with a K-Cube brushed DC servo motor controller (KDC101, Thorlabs, UK), a high-precision rotation mount (PR01/M, Thorlabs, UK), a large dual-axis goniometer (GNL20/M, Thorlabs, UK), an XYZ translation stage (LT3/M, Thorlabs, UK), a high-sensitivity CMOS camera (DCC3240C, Thorlabs, UK), a ring illumination lamp (Kern OBB-A6102, RS Components, UK), and a Nexus breadboard (B6090A, Thorlabs, UK). The data acquisition was facilitated by a DAQ (USB-6210 Bus-powered, National Instruments Ltd., UK) and a connected laptop (DELL, Austin, TX, USA). A Form 3B 3D printer (Formlabs, Somerville, MA, USA) with Grey Pro resin (Formlabs, Somerville, MA, USA) was utilised to fabricate the necessary parts for attaching the aforementioned components. A custom C# program was used to synchronise the stepper motor insertion with force measurements and video recording.

The one-axis sensor was attached to the motorised translation stage with a custom adapter to facilitate the insertion movement. The si*x*-axis sensor was attached to the dual-axis goniometer, located on the top of the rotation mount and the XYZ translation stage. The camera and the ring light were attached above the si*x*-axis sensor to illuminate the model correctly to observe the implant behaviour during the insertion (see Figure 1).

A practice cochlear implant electrode (Cochlear Slim Straight CI422, Cochlear Europe Ltd., UK) was attached to the one-axis sensor, and a 3D-printed artificial ST model was connected to the si*x*-axis sensor. A 1% solution of sodium dodecyl sulfate (SDS, Sigma Aldrich) in distilled water was injected into the model prior to the insertion for lubricating the lumen of the model [56,57]. The insertion speed, facilitated by the motorised translation stage, was set to 0.5 mm/s, and the insertion depth from the artificial model opening was set to 20 mm (only “small” and “tighter spiral” models were inserted to 17 mm). After the full insertion, a 5 s long pause was introduced, and then the electrode was retracted. Each model was implanted ten times combined over two identical CIs that were re-straightened by hand after every insertion.

To eliminate bubbles, the ST model was periodically filled with solution up until a point where no leakage of the fluid would occur due to surface tension at the ST basal opening.

### 2.6. Fitting of Insertion Forces to a Capstan Model

#### Capstan Model

It has been shown before [58,59] that the forces on the implant can be modelled similarly to a classical Capstan problem. The Capstan problem is a statics problem—Appendix A (left)—encountered when attempting to pull a rope around a rigid bollard. For a non-elastic, flexible, thin line on the verge of sliding around a rigid bollard, the problem can be modelled as
(2)$$T_{1}=T_{2}e^{\mu \theta }$$
where *T*_2_ is the load held by the restraining for *T*_1_, *θ* is the total angle subtended by the contact region of the rope, and *μ* is the coefficient of friction. Notably, the Capstan equation acts as a “force multiplier”, with the ratio between T_2_ and T_1_ being fixed for a given position on the verge of sliding. The exponential nature of this relationship is such that theoretically, for a coefficient of friction of 0.7, approximately that of steel on steel, a 1 kg restraining force would be capable of holding over 3.5 million tonnes with only 5 full turns.

In our case, the Capstan equation can be expressed as an overall force on the implant during insertion, *F_Implant_(θ)*, being related to the angular insertion along the ST wall, *θ*, according to:(3)$$F_{Implant}\left(\theta \right)=F_{tip}e^{\mu^{\prime }\theta }$$
where *F_tip_* is the tip force of the electrode, $\mu^{\prime }$ is the exponential coefficient that is linearly correlated to the coefficient of friction but includes other factors, including surface roughness and the spiral nature of the ST. Note that *θ*, in this case, is related to the angle relative to the initial contact point of the CI with the ST lateral wall rather than to the round window, measured in degrees.

When fitting the exponential growth of the insertion force exhibited on the implant with respect to insertion angle, *F_tip_* was fixed, as the tip force during initial contact between the CI and ST wall was observed to be very similar for all insertions within each experiment.

### 2.7. Statistical Analysis

MATLAB (Mathworks) ANOVA 1 with Multcompare function was used to study the statistical significance of exponential coefficients between the measurements. Data were found significant if *p* < 0.05. Each condition was replicated *n* = 5 times for two separate, but identical, Cochlear Slim straight electrodes for a total of 10 experimental repeats for each ST model.

## 3. Results

### 3.1. Insertion Setup with Accurate Scala Tympani Model

A workflow for creating the 3D printable CAD models of the ST was generated (Figure 1A), which included the characterisation of the micro-CT segmented cochlea and the manipulation of the ST shape before generating an STL file suitable for printing. The 3D-printed ST model and cochlear implant were secured to a six-axis and one-axis force sensor, respectively, to monitor the forces through the insertion (Figure 1B).

Using digital light processing (DLP) 3D printing, it was possible to produce highly accurate 3D models of the scala tympani cavity (Figure 2). Furthermore, through the addition of an acrylic coating after the standard post-processing on the inside and outside surfaces of the model, it was possible to significantly improve the transparency of the models, as seen in Figure 2A. The accuracy of these 3D prints was validated using a nominal–actual analysis to quantify the surface deviation of the 3D-printed ST with the original STL CAD file. This determined that 90% of the surface was within 32.1 µm of the original file, with the highest deviation occurring at the top surface of the basal and apical ends of the ST (Figure 2B). The localised deviation at the top surface of the ST is likely due to the printing of a free-standing surface without support structures. However, as the CI will not be in contact with these regions, they do not influence the CI insertion.

### 3.2. Influence of Overall Size on Insertion Force

Although some studies have found some relation between the overall size of the cochlea and the CI insertion force [46], as well as residual hearing preservation [60], a systematic study into the force dependence on size has not been previously conducted.

This study measured the forces exerted on both the implant and the cochlea. A six-axis force sensor provided the reactive force of the implant insertion in the *x*, *y*, and *z* axes (as depicted in Figure 1) which correspond to the forces in the direction of the implant insertion and perpendicular on the horizontal and vertical axes, respectively, as well as the torque around these axes. The overall force on the implant shows good agreement with the reaction force measured on the implant (R^2^ = 0.999), which acts as a good cross-validation of the two independent sensors (Appendix A). As expected, the overall force on the cochlea is dominated by the force in the direction of the CI insertion (along the *x*-axis), and the force along the perpendicular directions is approximately 10% of the magnitude of that primary force; see Appendix A.

As seen in Figure 3, the insertion force on the implant increased exponentially with the depth of the insertion. Additionally, the increase in this force was highly dependent on the size of the model, where a 10% increase or decrease in the overall volume (respectively, for the “large” and “small” models) of the model significantly impacts the insertion force. However, when normalising these profiles to the angular insertion depth rather than the length of the electrode inserted, the profiles overlap. This is as predicted by the Capstan model (described in Section 2.6) and is based on the perhaps unintuitive fact that the friction force is independent from the contact area between sliding objects and depends only on the total normal force and the coefficient of friction. For instance, in a “large” model, a longer length of the CI is in contact with the cochlear wall for a given angle when compared to a “small” model, but this just distributes the same overall normal force on a larger area. However, this suggests that there would be higher local stresses in a smaller cochlea due to the same overall force being distributed along a smaller contact area.

It should be noted that the “small” model was inserted only to a 17 mm insertion distance to preserve the structural integrity of the implant as a deeper insertion might damage the implant and change the forthcoming measurements.

The tip force (F_tip_) was determined as the force to bend the CI tip during the initial contact between the CI and the cochlear wall, at 100° depth relative to the round window. This remained consistent (at 3.00 ± 0.17 mN) between the different conditions and was fixed in fitting the Capstan model (Equation (3)) to the force exerted on the implant. The fitting of the force profile to the exponential Capstan model then determined the exponential coefficient μ’ (see Appendix A for the R^2^ error of the fitting and Appendix A for an example of the fitting). The exponential coefficients were not significantly different between the samples, suggesting that the insertion force is related to the angle of the CI insertion rather than the overall length of the CI in contact with the ST wall.

As the overall size influences many aspects of the cochlear geometry, as depicted in Table 1, a systematic variation in the different aspects of the cochlear geometry and their effect on the cochlear implant insertion force was conducted. These three main factors included (1) the vertical trajectory of the ST, (2) the horizontal trajectory of the ST (i.e., curvature), and (3) the cross-sectional area of the ST.

### 3.3. Influence of Scala Tympani Vertical Trajectory on Insertion Forces

Firstly, the manipulation of the ST vertical trajectory was conducted wherein the centreline of the ST cross-sections was unaltered except for their vertical position, as depicted in Figure 4A. This included producing a “flat” model where the centreline of all the cross-sections lay along the same x-y plane. The non-planar models introduced a sinusoidal variation in the vertical trajectory in the first 270°, with conditions NP1 and NP2 having a consistent amplitude of 0.2 mm but a period of 270° and 135°, respectively. This replicates the “rollercoaster” vertical trajectories observed in several studies [51,61,62]. The overall vertical trajectory (or rising spiral) of the ST centreline did not have a significant effect on the insertion force when considering the flat model versus the original ascending model. However, an increased non-planarity (condition NP2) led to a small but statistically significant decrease (*p* = 0.024 relative to the “original” model) in the insertion force on the implant and along the *z*-axis of the model, whereas the decreased frequency of the non-planarity led to a slightly higher force along the *z*-axis.

### 3.4. Influence of ST Curvature on Insertion Forces

The curvature of the ST models was changed by adjusting the parameter influencing the curvature of the inner spiral of the cochlea (θ_2_), as seen in Figure 5A. This was conducted on flat models; therefore, only the curvature was influencing the force profiles. As the curvature affected the angular insertion of the implant, this was a significant factor in determining the total insertion force for a given length of the inserted CI. Once normalised for the angular insertion depth, the IF profiles of all three (“flat”, “loose” spiral, and “tighter” spiral models) models overlapped and there was no statistically significant difference in their exponential coefficients (*p* > 0.05; see Appendix A). Similar to the “small” model, the “tighter” spiral model was also inserted to only a 17 mm insertion distance to preserve the structural integrity of the CI.

### 3.5. Influence of ST Cross-Sectional Area on Insertion Forces

Finally, the effect of the ST cross-sectional area was investigated (see Figure 6). Typically, there is a decrease in the cross-sectional area with an angle as the ST tapers from the base to the apex (see Appendix A). However, in this experiment, this was compared to a uniform cross-section where the cross-section of 1 mm depth from the round window was used along the whole spiral. Similar to the curvature experiment, the vertical trajectory was controlled for in this experiment by comparing to a “flat” model. When comparing the uniform cross-section model (“flat—uniform CS”) to the tapered cross-section model (“flat”), the insertion force is seemingly much smaller for a given insertion distance. However, when normalising for the angular insertion depth, the forces overlap as with other alterations of the ST geometry. When comparing the average exponential coefficient in the growth of the force with respect to the angle, there is no statistically significant difference (*p* > 0.05; see Appendix A) between these models.

## 4. Discussion

### 4.1. Comparison with Previous Work

This study represents a thorough analysis of the different contributions of the selected geometrical features, namely the basal planarity, vertical trajectory, overall scaling, curvature, and cross-section area of the ST on the CI insertion. We have demonstrated a method for systematically manipulating the different features of the ST shape by taking the cross-sections of a single ST segmentation, changing their position, and reconstructing them into a 3D mesh. Although others have used a cross-section analysis to characterise the ST shape [63], none have reconstructed these cross-sections into a 3D structure to investigate their effect on physical properties.

As far as the authors are aware, the shape manipulation algorithm developed for this study is the first implementation of a generalised lofting function in MATLAB for arbitrary cross-section shapes. This algorithm performed more reliably for this task than the lofting functions in established 3D design software, such as Autodesk Fusion 360. Furthermore, using a nominal–actual analysis, it was determined that the reconstruction was highly accurate to the shape of the original CAD model of the ST (90% of the surface with < 7.24µm deviation), as seen in Appendix A. At the apex of the cochlea, some meshing errors could occur due to the tight curvature of the cochlea, although this region was not of interest for the CI insertions and was not included in the manipulated ST 3D prints. Note that in the “flat” models, the ST was cut off at the point where one turn of the cochlea would intersect another due to being on the same plane but would always be beyond the level of the full CI insertion.

Furthermore, this study demonstrates the fabrication of directly 3D-printed models with a transparent finish and validated accuracy (90% of the surface within <32 μm deviation; see Figure 2). In contrast, the previous studies have either employed scaling ratios of the ST to accommodate for mismatches in their 3D-printed models [45,46,47] or used direct casting, which results in models that combine all three scalae and which does not allow for flexibility in manipulating its shape [64].

### 4.2. Impact of ST Shape on Insertion Forces

Overall, it can be seen that the insertion force on the CI is determined by the angular insertion depth and is rather resilient to other factors. Although the overall volume affected several parameters, as detailed in Table 1, the changes in the force were accommodated for by controlling for the angular insertion depth rather than considering the length of the implant inserted. All the changes in the ST geometry did not cause a statistically significant difference in the force relative to the angle; this provides strong evidence for the Capstan model. The only exception is when a large non-planarity is added to the base where the implant trajectory may be altered to a point that it does not follow the Capstan model, as discussed below. As the force increases exponentially with an angular insertion depth, it is very sensitive to changes in the angle, which were confirmed manually using the videos of each insertion.

#### 4.2.1. Effect of ST Vertical Trajectory

When controlling for the vertical trajectory of the ST, the ascending portion of the cochlea did not affect the force when comparing the “flat” and “original” models, both in terms of the overall force and the force in the vertical direction, as seen in Figure 4.

Introducing a high non-planarity to the basal turn (as with NP2) led to a small statistically significant decrease in the force on the implant. This somewhat counterintuitive result may be due to the CI having less contact with the lateral wall as it travels through the centre of the cochlear lumen. NP2 also had a lower overall z-force. However, this may be due to the implant being in contact with both the top and bottom walls of the ST and the sum of the vertical forces cancelling each other out. The CI diameter relative to the ST cross-section is demonstrated in Appendix A. Although statistically significant, this rather extreme case of non-planarity only results in a small difference in the insertion force which will not likely be clinically significant.

A typical amplitude of the non-planarity and fixed angle of the insertion was used in this study, as the non-planarity can be highly dependent on the coordinate system used to define the vertical trajectory of the cochlea [51].

#### 4.2.2. Effect of Curvature

The effect of the ST curvature on the insertion force was accommodated for by controlling for the angular insertion depth. In this study, only θ_2_, which varied the curvature of the inner spiral, was altered and the basal turn of the ST remained unaffected. Therefore, the insertion forces were similar in this region. As with the small model, a full insertion was not possible with the tight ST models as there was a significant risk of kinking the CI at deeper insertion depths.

#### 4.2.3. Effect of Cross-Sectional Area

The cross-sectional area of the ST was determined to have a minimal effect on the CI insertion force. The “original” ST varies from 2.6 to 1.0 mm^2^ across the extent of the CI insertion, whereas the “uniform cross-sectional” model was fixed at 2.5 mm^2^, as seen in Appendix A. It is worth noting that the “uniform cross-section” ST represents a rather extreme difference in the cross-sectional area between the models, which is beyond anatomical variation. The alteration in the cross-sectional area in the volume-scaled models (as illustrated in Appendix A), however, does not demonstrate a significant influence on the force with respect to the angular insertion depth.

As predicted by the Capstan model, the insertion force is determined by the angular insertion depth of the CI into the ST. Therefore, this finding reinforces the fact that it is the CI contact with the wall that determines the force rather than the overall space within the ST. At the depths inserted in this study, the cross-sectional area and the height of the lateral wall are significantly larger than the CI, as illustrated in Appendix A, respectively. For instance, at a 20 mm insertion, the height of the lateral wall in the original model varies from 1.6 to 0.9 mm (Appendix A), whereas the CI diameter varies from 0.6 to 0.3 mm from the base to the apex [60]. The size of the CI within the ST is illustrated more directly in Appendix A within a straightened ST. However, when the CI diameter would match the height of the ST, the insertion force and mechanical trauma are expected to increase significantly as the CI would be constrained by the top and bottom surfaces of the ST, deviating from the Capstan model.

### 4.3. Comparison with Surgical Approach

It should be noted that these experiments consisted of an insertion through a scala tympani with a fully open base rather than through a simulated round window or cochleostomy approach. Although not exactly the clinical approach, the round window anatomy can be very variable [65] and alters the angle of approach for the insertion. Therefore, by having a consistent insertion trajectory with an open base, it was possible to determine the influence of the ST size and shape on the insertion forces. This allowed the systematic determination of the contributors to the insertion force due to the ST shape. Future studies will focus on the angle approach of the CI insertion and the influence of many different segmentations of the cochlea and surgical approach rather than manipulating a single cochlea shape.

The amplitude range of the insertion forces measured in this study (~50–200 mN) were within the range measured in the cadaveric specimen listed in the literature [66,67,68,69,70]. However, these forces strongly depend on the angular insertion depth, which is often not reported; the treatment of the cadaveric specimen (e.g., a reduction in the endosteum—the soft tissue covering the inside of the ST lumen); and other parameters that might affect the coefficient of friction. Hence, it is difficult to compare the data with the published studies. Furthermore, no studies found used a Cochlear Slim Straight electrode as used in this study, which makes comparisons to the existing literature with different implants difficult. This supports the need for reporting insertion forces as a function of the angular insertion to ensure a fair comparison between studies.

### 4.4. Impact of Vertical Forces

The vertical forces exerted on the ST are important as they present a risk of damaging the basilar membrane and organ of Corti structures that are crucial in providing residual acoustic hearing. Therefore, measuring the effect of the force on the vertical *z*-axis could help determine the conditions of the increased risk of the basilar membrane damage and CI translocation between the scala, which can occur in up to 20% of lateral wall electrode implantations [71]. In our results, the spatial frequency of the variation in the non-planarity of the basal turn seemed to have differing effects on the insertion force. However, there was a significant variation in the force measured, as the range of the forces was reaching the limit of our sensor (a sensitivity of 1.5 mN for the *z*-axis). The vertical forces measured within this study are significantly lower (<5 mN) than those measured to rupture the partition, ranging from 42 to 122 mN [72], which included the bony osseous spiral lamina as well as the basilar membrane. However, the scalar translocation will largely depend on the localised stress applied to the cochlear partition, with the basilar membrane being significantly less stiff than the bony osseous spiral lamina and, therefore, being damaged at much lower forces.

### 4.5. Stress Relaxation of CI

Another factor that is related to the overall insertion forces is the elastic stress held in the CI, which can cause the CI to extrude due to stress relaxation. Due to the stepper motor-assisted insertion, a force relaxation could be observed when the CI was held in position at maximum insertion. The ratio of the force at a fixed distance to the maximum force was consistent across the conditions with a median value of 0.69, except for the “small” and “tight” models where a full insertion could not be achieved and therefore not fully comparable, and the results were more valid (see Appendix A). This is likely related to the inherent elasticity of the implant. This elasticity may vary across implant brands; hence, the same CI brand was used throughout this study to be consistent and eliminate the variability due to the implant mechanical properties. However, it was shown that there was no significant variability in the insertion force on the same model with repeated insertion (see Appendix A).

### 4.6. Consequences of Capstan Model

The basic Capstan equation has been used with significant success to understand the observed exponential behaviour of the cochlea insertion forces [59]. There are two particularly unintuitive observations, however, that have not been made.

The first consideration is that, for portions of the implant in contact with the ST wall, the bending stiffness does not affect the forces in that region. To see this, remember that
(4)M=EIκ
where M is the bending moment, E is the elastic modulus, I is the second moment of the area, and κ is the local curvature. It can be seen from the equilibrium conditions—Equation *(*S1)—that this term has no effect on the system solution as dM is zero for the locations of constant curvature. This counterintuitive fact was first noticed by Stuart et al. [73] for the classical Capstan problem and suggests that cochlea implant stiffness is not necessarily a limiting factor in the design. This comes with two major caveats, however.

Firstly, the bending moment does have a significant effect on the non-contact regions, such as at the base of the implant, and a stiffer implant may require a lateral constraint within a supportive stiff sheath.

Secondly, the bending moment does change which parts of the implant may be in contact. If the local tension/shear forces are not sufficient to hold the implant against the ST lateral wall, the forces will change.

Taken together, this suggests that the optimum implant stiffness profile is for a “pyramid of stiffness”, chosen to always be less than required to pull the implant away from the wall but great enough to maximise the steering control.

Nevertheless, the second consideration is just as significant: the angular insertion depth, coefficient of friction, and tip forces are the only significant factors affecting the implant forces. Features such as the ST size, flatness, and profile are only minor in their impact. Although these considerations would need to be directly investigated in a separate study with implants of varying stiffness, the fact that the ST curvature does not affect the insertion force suggests that the bending of the implant does not contribute significantly to the overall force. Particularly surprising is that the spiral geometry makes no difference at all in the model relative to the classical circular geometry used for a Capstan model. This suggests that the majority of the refinement effort in implant design should target the tip profiles and developing materials with low coefficients of friction.

### 4.7. Limitations of This Study

Although this study represents one of the more detailed studies of cochlear implant forces to date, there are still limitations to this setup. The conclusions of the Capstan model and overall forces on the cochlea do not let us investigate the local stresses on the cochlea and the identification of the local “hotspots” which could lead to localised insertion trauma. Therefore, there is a need for high-density force sensors that could be placed along the cochlea that could measure these localised forces. For instance, the buckling of the implant may push on the top and bottom surface of the ST and, therefore, cancel out forces measured with this setup.

## 5. Conclusions

In conclusion, after studying the parameters determining the CI insertion force, it is clear that accommodating for angular insertion depths accounts for most of the variation between the different ST geometries. Although the spatial frequency in the vertical trajectory of the basal turn may have a statistically significant effect on the insertion force, its small influence is unlikely to have a significant effect in surgery. These observations are summarised in Table 2.

This is promising in the pre-surgical planning of a CI insertion as even a basic analysis of the cochlear shape could feed into a predictive model of the insertion force and inform the decision of which CI and approach to use for a particular patient. Common measures such as the cochlear duct length and number of turns could be used to determine this angular insertion depth-to-distance relationship. Furthermore, to reduce insertion trauma, surgeons should consider implanting a CI to the same angular insertion depth rather than to a certain length of the implant. However, this comes with a trade-off between reducing trauma and achieving optimal CI electrode positioning to achieve effective neural stimulation. Additionally, the considerations within this paper relate to conventional straight electrodes that are positioned along the lateral wall rather than pre-curved electrodes which rely on the pre-tension to achieve a perimodiolar positioning.

By appreciating the consequences of the Capstan model that the tip force and coefficient of friction are the major determinants of the insertion force for a given angular insertion depth, it is clear that developing new CI tip designs and surface coatings to reduce friction will likely be most effective in reducing insertion trauma. Furthermore, the Capstan model shows that an increased stiffness of the implants may not increase the insertion forces so long as they do not affect the implant following the lateral wall.

By combining these insights to further understand the intracochlear forces during insertion, it may be possible to improve the CI insertion to provide an optimal electrical stimulation while minimising the trauma. This could improve CI users’ outcomes by retaining more of their residual hearing that provides acoustic cues to improve their hearing. Additionally, by reducing the risks of the CI insertion, it could be possible to widen the eligibility of CIs to include those with less severe hearing loss to provide these benefits to a much wider patient population.

## Figures and Tables

**Figure 1 biosensors-12-00999-f001:**
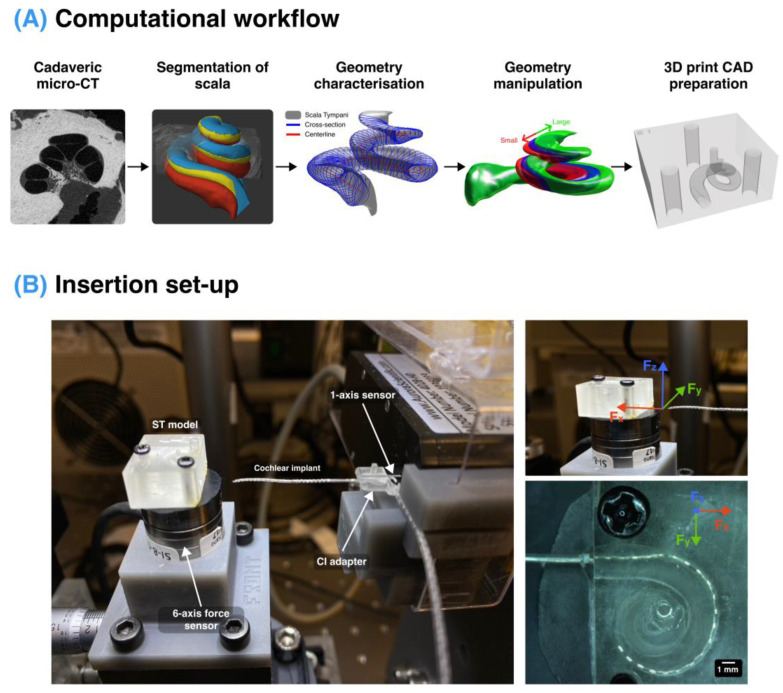
Computational workflow and the custom insertion setup. (**A**) A cadaveric specimen was micro-CT scanned and segmented using Stradview software. The 3D geometry was then characterised in terms of its spiral trajectory and cross-sections using a custom MATLAB script which was used to manipulate the geometry according to each experiment. This generated a 3D file of the ST which was prepared for 3D printing using another custom MATLAB script. (**B**) Insertion setup consists of transparent 3D-printed ST models affixed with screws to a six-axis force sensor placed on a six-axis positioning stage and a CI secured with 3D printed adapters to a one-axis force sensor placed on a stepper motor that moved the CI into the ST model at a defined speed. The setup measures the force on the ST model in the direction of implantation (x), perpendicular in the left and right directions (y), and vertically (z), shown on the right. The one-axis force sensor measured the reaction force on the implant.

**Figure 2 biosensors-12-00999-f002:**
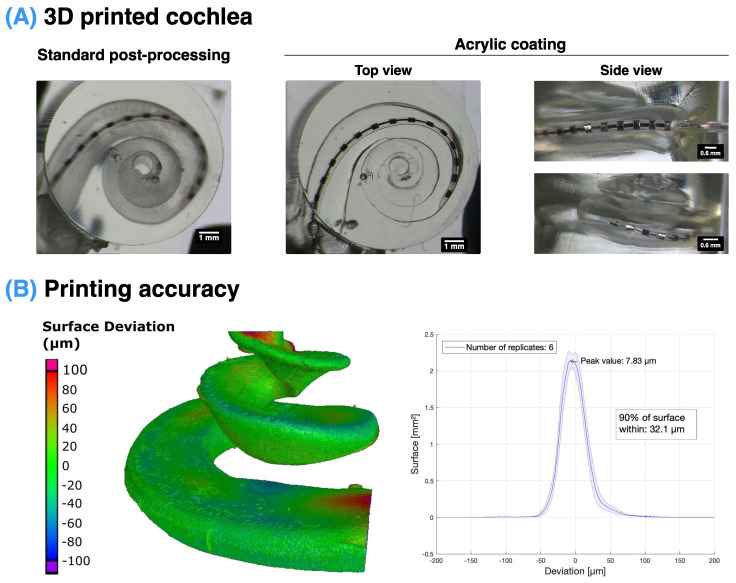
(**A**) Implantation of a CI in ST models with manufacturer-recommended post-processing (**left**) and an additional acrylic coating (**right**). This demonstrates the clear imaging of the CI electrodes, vertically and horizontally, to investigate CI positioning and validate the angular insertion depth. (**B**) Quantification of 3D-printing accuracy using nominal–actual analysis demonstrating the 3D map (**left**) and histogram of surface deviation (**right**) compared to the original CAD file used for printing.

**Figure 3 biosensors-12-00999-f003:**
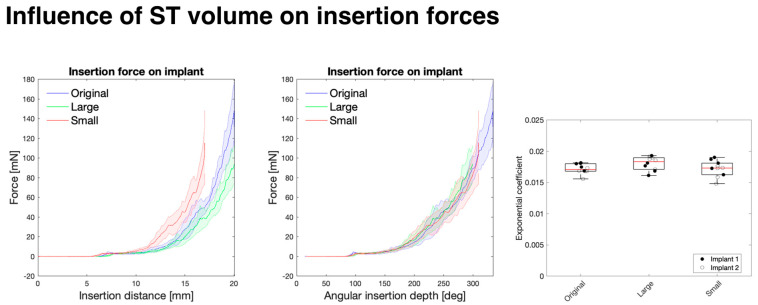
Influence of ST overall size on insertion forces. A volumetric scaling of the ST model was conducted to produce a “large” (110% the volume of the “original”) and ”small” (90% of the “original” model) models. Whereas a significant difference is seen when plotting force exerted on the CI (mean-solid line; shaded area-standard deviation) with respect to insertion distance (**left**), these force profiles overlap when plotting relative to angular insertion depth of the round window (**middle**). The exponential coefficients (**right**) of the Capstan model fitting to the force on the implant with respect to the angular insertion depth illustrate no significant difference (*p* > 0.05) between the insertion force profiles. Boxplot: red line-median, box-interquartile range (n = 10 replicates combined over N = 2 implants).

**Figure 4 biosensors-12-00999-f004:**
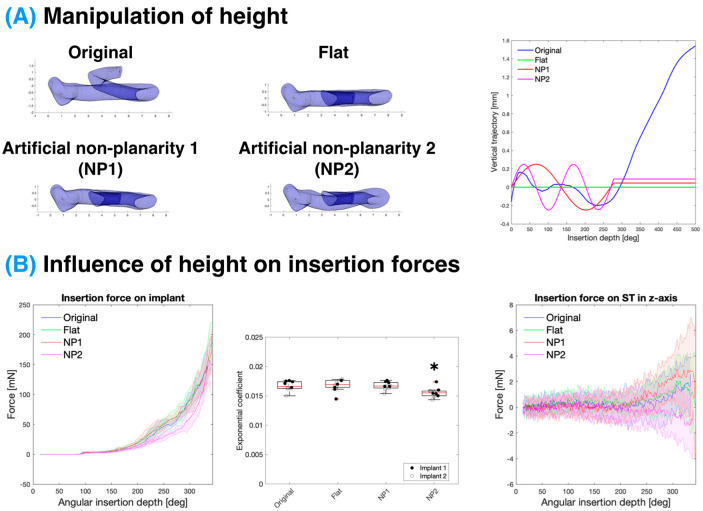
Manipulation of the vertical trajectory of the ST and its influence on CI insertion forces. (**A**) Representation of the 3D geometry (**left**) and the centreline z-component relative to the basal plane (**right**) of the different ST models, only varying in the vertical trajectories of their centrelines; these consist of the original geometry (cut off at 500° for comparison), a flat model where all centrelines are on the same horizontal plane, and two artificial non-planarities (NP1 and NP2). (**B**) Insertion force on the implant with respect to angular insertion depth (**left**; solid line-mean, shaded area-standard deviation) and respective fitting of the Capstan model exponential coefficient (**middle**; red line-median, box-interquartile range) for models with different vertical trajectories. Data represent n = 10 replicates per condition over N = 2 implants. Only NP2 showed a statistically significant difference (*p* values of 0.024, 0.010, and 0.020 compared to “original”, “flat”, and NP1 models, respectively). Vertical forces, along the *z*-axis, on the ST model due to CI insertion (**right**). Note, altering the vertical trajectory made little difference to the angular insertion depth per mm of length.

**Figure 5 biosensors-12-00999-f005:**
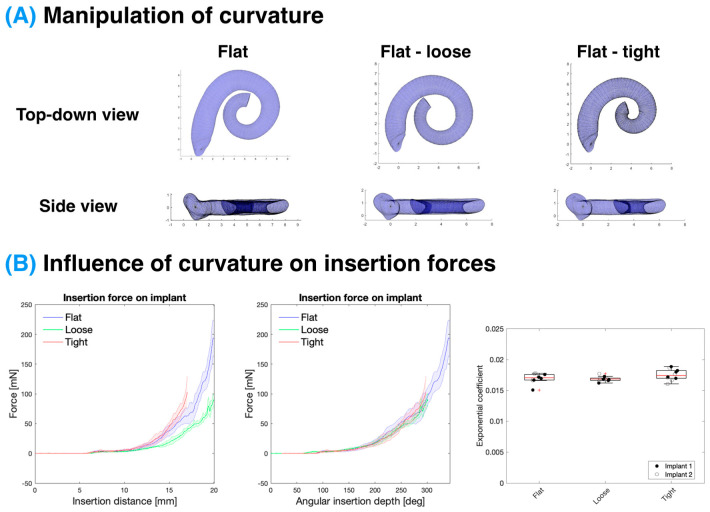
Influence of ST curvature on the CI insertion force. (**A**) Representations of the shape manipulation of the ST models. (**B**) Insertion force experienced by the implant for models with altered curvatures with respect to insertion distance (**left**) and angular insertion depth (**middle**; solid line-mean, shaded area-standard deviation). Statistical analysis of the exponential coefficients, which are acquired by fitting the insertion force profiles, shows no statistical significance between the models (*p* > 0.05; boxplot: red line-median, box-interquartile range). Data represent n = 10 replicates per condition combined over N = 2 implants.

**Figure 6 biosensors-12-00999-f006:**
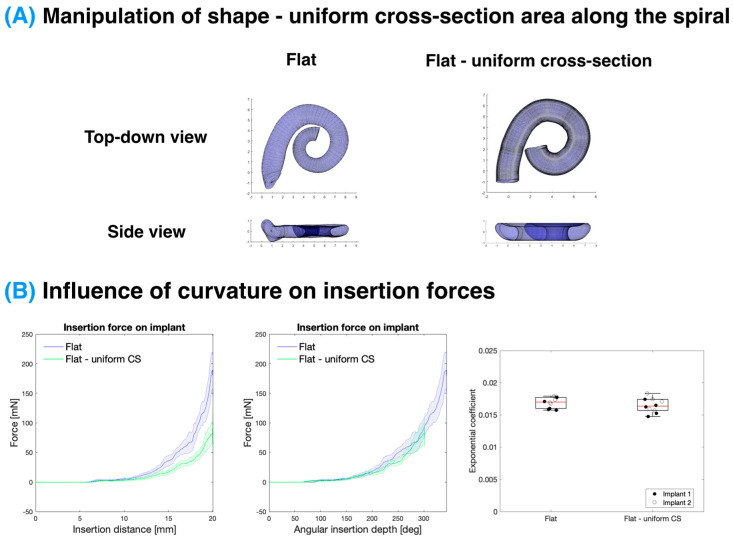
Influence of ST cross-sectional area on insertion forces. (**A**) Manipulation of the ST geometry demonstrating the tapered cross-sectional area of the “flat” model versus the “flat-uniform cross-sectional area” model where the cross-section at 1 mm depth from the round window was used along the same horizontal trajectory. (**B**) Force exerted on the CI during insertion as a function of insertion distance (**left**) and angular insertion depth (**middle**; solid line-mean, shaded area-standard deviation) along with the exponential coefficient of the fitting of the force profiles (**right**; red line-median, box-interquartile range) demonstrates no significant effect (*p* > 0.05) of the uniform cross-section area on insertion force. Data represent n = 10 replicates per condition combined over N = 2 implants.

**Table 1 biosensors-12-00999-t001:** Summary of manipulation of ST size/shape and its impact on ST height of the lateral wall, and trajectory of the CI in the vertical axis and horizontal plane.

		IMPACT ON:
		Height of LW	Trajectory in Vertical Axis	Trajectory in Horizontal Plane
**MANIPULATION OF MODEL:**	Volume	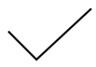	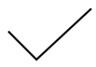	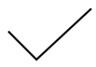
Basal planarity and rising spiral	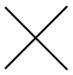	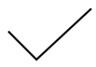	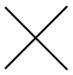
Curvature	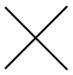	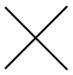	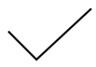
Cross-section area	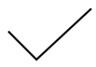	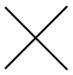	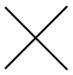

**Table 2 biosensors-12-00999-t002:** Summary of the effect of ST shape manipulation on cochlear insertion forces.

		Impact on Insertion Force with Respect to Angular Insertion Depth
**MANIPULATION OF MODEL:**	Volume scaling	No statistically significant difference (*p* > 0.05)
	Overall vertical trajectory/ rising spiral	No statistically significant difference (*p* > 0.05)
	Basal turn non-planarity	Higher non-planarity may decrease insertion force due to less contact (*p* > 0.01)
	Curvature	No statistically significant difference (*p* > 0.05)
	Cross-section area	No statistically significant difference (*p* > 0.05)

## Data Availability

Available upon request.

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
