# Peer review of "Impact of Scala Tympani Geometry on Insertion Forces during Implantation"

_biosensors, 2022, doi:10.3390/bios12110999_

Round 1
Reviewer 1 Report
General comments
The authors used 3D-printed transparent resin ST models to investigated CI insertion forces caused by the CI-insertion process. Geometrical parameters of the ST were gained by microCT investigation of cadaver. The mathematical model allowed the authors to reprint the resin-ST model using modified geometrical parameters and to test the effect on the insertion force.
Although, the topic is significant, figures are clear and attractive, there are some significant points to consider.
Major comments
The authors repeated experiments only three times to insert the same CI in the 3D-printed models (line 209-210). Statistical analysis made on the exponential coefficient gained by fitting the data often showed almost or almost not significant differences. Larger number of performed tests would have significantly increased the possibility that the presented geometry alterations resulted statistically significant changes of the exponential coefficient also in the cases where it was found not be significant using n =3.
Legends of the figures should contain all the information about the given figure. Figures and its legends should be understandable without additional explanation. It is not the case.
The authors used the Capstan equation to gain information from the measured insertion force by fitting the data. It would have been welcomed to see in some figure the result of the fit to be demonstrated. The deviation of the applied model, that is obvious at around the insertion depth of 250° is not discussed.
Figure numbering is misleading. That could lead the authors to leave the Figure A 4 in without description or even link in the manuscript. The Figure 6 is also not linked/mentioned in the manuscript. In case the authors find results important for the manuscript it is recommended to describe all of them in the “Results”. Figures labelled with “A” (Figure A 1 – A 10) could be presented as supplementary material.
The authors demonstrate in the “Results” that the investigated parameter was not altered significantly in the flat model compared to the original model. However, the authors in the “Discussion” demonstrate that the same investigated parameter investigating the flat and the original models are different and the difference is statistically significant. Such “mistake” does not help the understanding of the manuscript.
Minor comments
Line 2: What explain the plural form of the force? Although, it is not defined the insertion force is most probably the force that is applied to push the CI into the cochlea. That force has three the x, the y and the z components.
Line 169-171: Was the access hole closed prior to insertion of CI? Did the leakage of the ST fluid influence the insertion force?
Line 172: What was the accuracy/resolution of the applied 3D printer?
Line 180: Most probably the coating fluid was applied and not the coating was removed after the application. Please correct the sentence!
Line 181: What was the function of Pluronic, how did it work?
Line 201: What was the function of the camera? What type of camera was used?
Line 205: What was the purpose to use 1% sodium dodecyl sulfate instead artificial perilymph to fill the models? How could the difference of the fluids influence the measured insertion force?
Line 209-210: It is stated that each model was implanted three times, however there are more than three data points are plotted in the Figure A 7. Please explain the reason!
Line 231: How does the type of coating influence the exponential coefficient? Has it been tested using different type of coating? Has already been tested to modify the surface roughness e.g. by changing the printing resolution or by using different coating? What could be the correlation between the exponential coefficients of the real and the model cochleae?
Line 240: The paragraph “Statistical Analysis” is not complete. It is not required to put all the statistical data in the text, but the significant data have to be presented in the manuscript, including P, t and n values, e.g. in the legend of the figures.
Line 257: What does the “fully” mean?
Line 275: The nominal-actual analysis proved that the 3D-printed model is very similar to the CAD file. The question whether the same is true for the original ST and the CAD file? How likely that during the CAD-file generation deviations from the original tissue were made? How likely it is that roughness of the ST in the cadaver could not be implemented in the CAD file?
Line 311: What does the “largely” mean? It is it either related or not related, please decide.
Line 313: “The” instead of “This”.
Line 319: What do the “samples” mean? Please explicit state what sort of samples they are.
Line 346-350: It is recommended to indicate the mean and the SD values including the result of the statistical test in the figure legend.
Line 348-349: “led to a slight decrease… …and along the z-axis of the model” – please show the data and the result of the statistical analysis! What does the “slightly higher” mean? Was the difference statistically significant? Statistically significant deviations have to be documented in the manuscript.
Line 370-371: The influence of the curvature was investigated on flat models. Although, the flattening alone did not significantly influence the exponential coefficient (Figure 4(B) middle), the flattening itself was already modification on the original form. Why not the same test strategy was chosen, as in the case of the vertical trajectory influence, where the not uncoiled models were tested. In case, the not-flattened tight and the not-flattened loose models have not been created, the possible expectations could be discussed.
Line 388: The aim of this paragraph was most probably to describe the data demonstrated in the Figure 6, however there is no link in the entire manuscript to that figure. The influence of the ST cross-section area was tested, similarly to the curvature tests, on flattened models. My comment is similar to the curvature tests; therefore, a possible effect could be discussed in case of not-flattened models.
Line 402: In case the “Large” and the “Small” models are the same as described previously, please refer to them in the legend.
Line 406-407: Please list here the “selected geometrical features”, that were studied. The geometrical features of the CI are most probably not included.
Line 417-418: In case that sentence is aimed to be a statement that has to be presented in the “Results”.
Line 420: What does the “original ST” mean? Do the authors under the “original ST” the original ST model mean? Please rewrite more clearly!
Line 433: Why was the “transparent finish” significant?
Line 470-471: “remained largely unaffected”: What does the largely unaffected mean? Please indicate the result of the statistical analysis to be able to decide whether it was affected or not!
Line 484: That has to be discussed here and not in the “Results”.
Line 503: It would be reasonable to discuss the amplitude range of the insertion forces measured in the current study compared to that occur during manual insertion.
Line 520-521: Give a link to the results.
Line 523: Please indicate the range of forces measured in the study!
Line 526: A rough calculation could be made to estimate the possible local force based on the data of the manuscript.
Line 538-540: Please give a link to the results where the effect of repeated insertion was demonstrated. Please also show that the variability was statistically not significant.
Line 550-551: There are typos in the sentence. Bracket has to be closed and there is no Equation 5 in the manuscript.
Line 557: What is the capillary tube?
Line 567: How the authors define “spiral geometry”, because the Figures 4, 5, 6, and A 7 demonstrate statistically significant changes on the insertion force by modifying the geometry of the 3D-printed models.
The “References” requires corrections. The references #2 (line 647), #13 (line 673), #44 (line 750-751), #49 (line 761-762), #52 (line 767), #53 (line 768) and #56 (line 773) are not complete.
Figure 1: Scale bar could be applied to panels A (left) and B. What was the function of the brown wire on panel B (left)?
Figure A 2: How was the total insertion force on ST calculated?
Figure 2: Pease add scale bars to the panel A!
Figure A 3: The legend of a figure has to contain all the information related to the given figure. Please introduce the meaning of “Original”, “Large” and “Small” at the first appearance, therefore here. Actually, the task of this figure is to compare the total force with the x, y and z components, therefore I would recommend to remove the “Large” and “Small” data from this figure. Without description it could only be speculated that most probably the mean and SD values of the three test are represented in all the three cases.
Figure 3: What are the horizontal bars, pulled through the red dots, represent in the panel right? Please, add the P value of the statistical test and declare the criteria of significance in the methods. How could it be estimated that the initial condition, namely the curvature of the CI, was always restored to the original condition.
Figure 4: Why were the NP1 and NP2 models calculated not deeper than insertion depth of ~280°, however the “original” was made up to 500 (Figure 4, panel A right)? It could be expected that vertical alteration of the model would affect not the force components in the x-y plane, but in z direction. Therefore, it would be interesting to see the exponential coefficients derived from the z-axis forces.
Figure 6: The data demonstrated in the Figure 4 and Figure 5, gained by using flat models, are different to the data showed in the Figure 6! Where does the difference come from? That change most probably affected the result of the statistical analysis. Please describe more accurately if this flat model is different to the previous one! In the description of panel B only panels “left”, which related to the middle, and “right” are mentioned. In case the real left is not important remove that panel or built a link.
Figure A 4: Where does the Figure A 4 belong to? In case the authors do not find it relevant to mention it in the manuscript, please delete it.
Figure A 7: What is the difference between the data in this figure and in the Figure 4 (Original vs. Flat)? In the Figure 4 only data n = 3 are presented, however here n > 3. It is also demonstrated in the Figure 4 that the difference between original and flat models is not significant, however in Figure A 7 demonstrated in the “Discussion” the difference between the original and flat models is statistically significant. The difference is not clearly described in the manuscript.
Figure A 8: It would be informative, if possible, to indicate in the figure the maximal insertion angle of a 20-mm long CI.
Author Response
| Major comments | Reply |
| The authors repeated experiments only three times to insert the same CI in the 3D-printed models (line 209-210). Statistical analysis made on the exponential coefficient gained by fitting the data often showed almost or almost not significant differences. Larger number of performed tests would have significantly increased the possibility that the presented geometry alterations resulted statistically significant changes of the exponential coefficient also in the cases where it was found not be significant using n =3. | We've now repeated 5 replicates on 2 separate implants (same brand and model) to improve the statistics and our confidence in the results. As these are real implants from CI companies they are relatively precious therefore practically we can only repeat on 2 separate implants. Furthermore, we would not want to do much more insertions with the same implants as not to affect the integrity of the implant itself. However, in the case of these 5 replicates no significant difference was seen across the experiment indicating the implant was not significantly affected by the repeated insertions. |
| Legends of the figures should contain all the information about the given figure. Figures and its legends should be understandable without additional explanation. It is not the case. | Figure captions have been updated to improve their readability |
| The authors used the Capstan equation to gain information from the measured insertion force by fitting the data. It would have been welcomed to see in some figure the result of the fit to be demonstrated. The deviation of the applied model, that is obvious at around the insertion depth of 250° is not discussed. | With the repeated measures, that deviation near 250 has been resolved and was likely due to relaxation of the implant due to slight kinking of the electrode which is very difficult to avoid. |
| Figure numbering is misleading. That could lead the authors to leave the Figure A 4 in without description or even link in the manuscript. The Figure 6 is also not linked/mentioned in the manuscript. In case the authors find results important for the manuscript it is recommended to describe all of them in the “Results”. Figures labelled with “A” (Figure A 1 – A 10) could be presented as supplementary material. | Figure labelling has been updated for the supplementary information. The figure layout was altered where our supplementary figures which we had placed at the end of the manuscript were added throughout the main manuscript and therefore led to this confusing figure numbering. |
| The authors demonstrate in the “Results” that the investigated parameter was not altered significantly in the flat model compared to the original model. However, the authors in the “Discussion” demonstrate that the same investigated parameter investigating the flat and the original models are different and the difference is statistically significant. Such “mistake” does not help the understanding of the manuscript. | This was due to the controls being repeated for each singular experiment (such as influence of ST volume, height, curvature, and shape) and a comparison was made between these different controls. However, we entirely agree that this may be misleading and have repeated the whole experiment with an equal amount of repeats. |
| Minor comments | |
| Line 2: What explain the plural form of the force? Although, it is not defined the insertion force is most probably the force that is applied to push the CI into the cochlea. That force has three the x, the y and the z components. | The forces refer to both the x,y,z components of the force that could be measured on the cochlea models themselves as well as the reactive force measured on the implant. |
| Line 169-171: Was the access hole closed prior to insertion of CI? Did the leakage of the ST fluid influence the insertion force? | Comment added to clarify this at the end of section 2.5 (and address the elimination of bubbles). The access hole was not closed as not to affect insertion of the CI. We ensured that the ST wasn't overfilled so surface tension was enough to ensure the liquid remained in the ST, even throughout insertion. But the solution was topped up to accomodate for evaporation. |
| Line 172: What was the accuracy/resolution of the applied 3D printer? | 30 um. Which has been added to the method section 2.4. This was validated with nominal actual analysis (Figure 2 C) where the accuracy of the 3D print compared to the original model was 32 um. |
| Line 180: Most probably the coating fluid was applied and not the coating was removed after the application. Please correct the sentence! | This sentence has been clarified. The coating was injected and then removed with compressed air to leave only a thin layer on the walls. |
| Line 181: What was the function of Pluronic, how did it work? | Pluronic is a surfactant that is commonly used to increase the hydrophilicity and reduce the adhesion to surfaces. In this case it enabled the reduction of the friction to be more similar to the in vivo environment. |
| Line 201: What was the function of the camera? What type of camera was used? | The camera was used to acquire videos throughout the insertion using a Thorlabs CMOS camera specified in the methods (line 193). This was used |
| Line 205: What was the purpose to use 1% sodium dodecyl sulfate instead artificial perilymph to fill the models? How could the difference of the fluids influence the measured insertion force? | SDS was used to lubricate the lumen of the model prior to the insertion. Application of different fluids have an impact on the insertion forces (lubrication will lower insertion forces) as it affects the friction. A wide range of lubricants has been used in literature - we tried different concetrations of SDS and decided on 1% as a compromise of providing lubrication to bring the measurement within the clinical/cadaver range while not risking having a sopay film deposit on the implant which might accumulate with repeat insertions. |
| Line 209-210: It is stated that each model was implanted three times, however there are more than three data points are plotted in the Figure A 7. Please explain the reason! | The original and flat models were used as controls that were repeated for each separate experiment to eliminate any effect of drift in the sensor. Therefore, we acquired more datasets for these conditions as they were repeated in many of the experiments. However, we've now repeated conducted more repeats and are using an equal number of data points for the plotting and stats for all conditions throughout the manuscript. |
| Line 231: How does the type of coating influence the exponential coefficient? Has it been tested using different type of coating? Has already been tested to modify the surface roughness e.g. by changing the printing resolution or by using different coating? What could be the correlation between the exponential coefficients of the real and the model cochleae? | Questions regarding coating has been addressed in comments line 205. We tried SLA (with 25 μm resolution in XYZ) and DLP (with resolution 30 μm in XY, 30 μm in Z and 5 μm in Z) printers for the fabrication of the models. We received the best results (meaning the lowest deviation over 90% of the print surface) with DLP printer and 30 μm resolution in XYZ. 5 μm resolution in Z-axis had higher deviations as the clear resin is yet to be optimised for this 5μm resolution printing. We also explore using the resin itself as a coating treatment, however this resulted in higher deviations than the acrylic treatment. |
| Line 240: The paragraph “Statistical Analysis” is not complete. It is not required to put all the statistical data in the text, but the significant data have to be presented in the manuscript, including P, t and n values, e.g. in the legend of the figures. | Statistical data has been added in the text and summarised in table 2 and 3 |
| Line 257: What does the “fully” mean? | Added more detailed description of the characterisation |
| Line 275: The nominal-actual analysis proved that the 3D-printed model is very similar to the CAD file. The question whether the same is true for the original ST and the CAD file? How likely that during the CAD-file generation deviations from the original tissue were made? How likely it is that roughness of the ST in the cadaver could not be implemented in the CAD file? | The resolution of the microCT is not high enough to resolve the surface roughness of the cochlea plus the 3D printing resolution won't be able to replicate this roughness. However, we suspect the frictional coefficient will be largely determined by the material properties rather than topography as the cochlear lining is quite smooth. This facilitates the need for coatings and SDS to make the insertion similar to cadaveric experiments. This is sufficient for the aims of this paper which looks at the relative changes due to geometric changes in the cochlea. |
| Line 311: What does the “largely” mean? It is it either related or not related, please decide. | Added further explanation at line 308 and following sentence. The insertion force is related to the normal force applied to the wall but there will be a higher local stress if this force is distributed on a smaller area. |
| Line 313: “The” instead of “This”. | Done |
| Line 319: What do the “samples” mean? Please explicit state what sort of samples they are. | Corrected. It was between the differently scaled models. |
| Line 346-350: It is recommended to indicate the mean and the SD values including the result of the statistical test in the figure legend. | This has been added. |
| Line 348-349: “led to a slight decrease… …and along the z-axis of the model” – please show the data and the result of the statistical analysis! What does the “slightly higher” mean? Was the difference statistically significant? Statistically significant deviations have to be documented in the manuscript. | Updated and clarified with new results |
| Line 370-371: The influence of the curvature was investigated on flat models. Although, the flattening alone did not significantly influence the exponential coefficient (Figure 4(B) middle), the flattening itself was already modification on the original form. Why not the same test strategy was chosen, as in the case of the vertical trajectory influence, where the not uncoiled models were tested. In case, the not-flattened tight and the not-flattened loose models have not been created, the possible expectations could be discussed. | To eliminate the compounding effect of different variables it was decided that the flat model would be used as a baseline control for following experiments as height would then be eliminated as a factor and the influence of curvature (for example) could be determined. |
| Line 388: The aim of this paragraph was most probably to describe the data demonstrated in the Figure 6, however there is no link in the entire manuscript to that figure. The influence of the ST cross-section area was tested, similarly to the curvature tests, on flattened models. My comment is similar to the curvature tests; therefore, a possible effect could be discussed in case of not-flattened models. | Reference to figure 6 added |
| Line 402: In case the “Large” and the “Small” models are the same as described previously, please refer to them in the legend. | Large and small models are the same throughout and represent volumetric scaling by 10% compared to the original ST anatomy. A section in the methods has been added to explicity describe their generation as well as clarification in the caption. |
| Line 406-407: Please list here the “selected geometrical features”, that were studied. The geometrical features of the CI are most probably not included. | This has been added to the manuscript. |
| Line 417-418: In case that sentence is aimed to be a statement that has to be presented in the “Results”. | This is described in section 3.2 of the results and further discussed in the discussion |
| Line 420: What does the “original ST” mean? Do the authors under the “original ST” the original ST model mean? Please rewrite more clearly! | Original ST refers to the original, unaltered, scala tympani model. However, in this context it is comparing the CAD file with the 3D print and we've clarified the text. |
| Line 433: Why was the “transparent finish” significant? | The transparent finish allowed us to monitor the electrode positions over time. This both allowed us to confirm that the trajectory of the implant was as expected. |
| Line 470-471: “remained largely unaffected”: What does the largely unaffected mean? Please indicate the result of the statistical analysis to be able to decide whether it was affected or not! | Updated with statistical significance of new results. |
| Line 484: That has to be discussed here and not in the “Results”. | Corrected. Reference to results not needed. |
| Line 503: It would be reasonable to discuss the amplitude range of the insertion forces measured in the current study compared to that occur during manual insertion. | The amplitude range of inseriton forces seems to be less relevant in this study as we focused on the overall impact of shape/size of ST on insertion forces. Hence the measured forces are rather relative than absolute. From the literature, measured insertion forces in cadavers are 20 - 200 mN; however, this depends strongly on the angular insertion depth which is often not stated (only the insertion distance is which is less relevant as we showed in this study). Following that, most of the studies we found used MED-EL electrode which are physically very different from Cochlear Slim Straight, therefore, the amplitude forces might be even less relevant. |
| Line 520-521: Give a link to the results. | Added an additional link to the curvature results. However, we acknowledge that the cannot directly claim that the stiffness will not affect insertion forces from the results in this study but think it's an important consideration from the Capstan model shown which should be a topic for further research. |
| Line 523: Please indicate the range of forces measured in the study! | It can be difficult to compare the forces between studies due to the exponential increase in force means that small changes in angular insertion depth can hugely change the force experienced. However, a comment on the force ranges has been added to section 4.4 |
| Line 526: A rough calculation could be made to estimate the possible local force based on the data of the manuscript. | The force distribution along the implant actually follows an inverse exponential to the force-angle relation seen. Therefore, the local stress on the implant exponentially increases from tip to being maximal at the base. |
| Line 538-540: Please give a link to the results where the effect of repeated insertion was demonstrated. Please also show that the variability was statistically not significant. | Example of repeats is demonstrated in supplementary figure S11. No significant trend was seen when checking exponential coefficients over insertions. |
| Line 550-551: There are typos in the sentence. Bracket has to be closed and there is no Equation 5 in the manuscript. | Corrected. Equation referred to is in supplementary info |
| Line 557: What is the capillary tube? | A capillary tube is a common glass tube used for different applications. Replaced with stiff sheath to make it more clear. |
| Line 567: How the authors define “spiral geometry”, because the Figures 4, 5, 6, and A 7 demonstrate statistically significant changes on the insertion force by modifying the geometry of the 3D-printed models. | We have clarified this sentence as this is referring to formulating the Capstan model with a spiral relative to a circle. I.e. using a spiral makes no difference when comparing things to a classical Capstan model in a circular geometry. |
| The “References” requires corrections. The references #2 (line 647), #13 (line 673), #44 (line 750-751), #49 (line 761-762), #52 (line 767), #53 (line 768) and #56 (line 773) are not complete. | This has been addressed. Citation Jacobson A, others(2021) was cited according to author's preferences posted at http://github.com/alecjacobson/gptoolbox . Citation geom3d is citing MATLAB package. Citation Dang K (2017) is a thesis. |
| Figure 1: Scale bar could be applied to panels A (left) and B. What was the function of the brown wire on panel B (left)? | Scale bar has been added. The brown wire is a hydrostatic pressure sensor which we did not use in this paper. The photo has been updated accordingly. |
| Figure A 2: How was the total insertion force on ST calculated? | The total insertion force was calculated following this equation: sqrt(Fx^2 + Fy^2 + Fz^2). This has been added to the figure captions. |
| Figure 2: Pease add scale bars to the panel A! | Scale bars have been added. |
| Figure A 3: The legend of a figure has to contain all the information related to the given figure. Please introduce the meaning of “Original”, “Large” and “Small” at the first appearance, therefore here. Actually, the task of this figure is to compare the total force with the x, y and z components, therefore I would recommend to remove the “Large” and “Small” data from this figure. Without description it could only be speculated that most probably the mean and SD values of the three test are represented in all the three cases. | This has been addressed. |
| Figure 3: What are the horizontal bars, pulled through the red dots, represent in the panel right? Please, add the P value of the statistical test and declare the criteria of significance in the methods. How could it be estimated that the initial condition, namely the curvature of the CI, was always restored to the original condition. | Boxplot: horizontal red lines represent median, box represents interquartile range (n=10 replicates combined over N=2 implants). This now is explicityly stated in each caption. As the insertion force profiles remained stable over 5 insertions per implant for each model which was randomly sampled to avoid bias across the experiment, we are confident that the intergrity of the implant remained consistent throughout the experiment. |
| Figure 4: Why were the NP1 and NP2 models calculated not deeper than insertion depth of ~280°, however the “original” was made up to 500 (Figure 4, panel A right)? It could be expected that vertical alteration of the model would affect not the force components in the x-y plane, but in z direction. Therefore, it would be interesting to see the exponential coefficients derived from the z-axis forces. | It has been shown by Gee et al that the variation of the vertical trajectory mainly occurs in the first 270 degrees before it then ascends therefore we decided to replicate this initial non-planarity with a sinusoidal variation. To compare with further experiments where the flat model was used as a control it was decided to make these models flat beyond 270 degrees rather than ascending. |
| Figure 6: The data demonstrated in the Figure 4 and Figure 5, gained by using flat models, are different to the data showed in the Figure 6! Where does the difference come from? That change most probably affected the result of the statistical analysis. Please describe more accurately if this flat model is different to the previous one! In the description of panel B only panels “left”, which related to the middle, and “right” are mentioned. In case the real left is not important remove that panel or built a link. | The difference was due to the flat model being remeasured as a control for each separate experiment. However, this has now been rectified with the new data of all 10 replicates on 2 different implants. Caption has been corrected. |
| Figure A 4: Where does the Figure A 4 belong to? In case the authors do not find it relevant to mention it in the manuscript, please delete it. | Figure S4 is referred to in section 3.2 which compares the volumetrically scaled ST models. |
| Figure A 7: What is the difference between the data in this figure and in the Figure 4 (Original vs. Flat)? In the Figure 4 only data n = 3 are presented, however here n > 3. It is also demonstrated in the Figure 4 that the difference between original and flat models is not significant, however in Figure A 7 demonstrated in the “Discussion” the difference between the original and flat models is statistically significant. The difference is not clearly described in the manuscript. | This is now discounted with the newly added repeats and statistical analysis. |
| Figure A 8: It would be informative, if possible, to indicate in the figure the maximal insertion angle of a 20-mm long CI. | Done |
Reviewer 2 Report
The paper uses geometric data of the human cochlea in order to model 3D printed experimental setups for measuring forces during electrode ("cochlear implant procedure") insertions. Model data are varied and the results show independence of all parameters save for the insertion depth/total angle. Thus a classical angular contact resistance model is likely to hold true (forces exponentially growing with angle).
headline 2.1 starts with a small letter.. micro-CT...
formulas 2 + 3 really should have the angle in radians, one might think degrees is Ok. The authors sure know this.
I think the paper is very good and I have no real concerns on its validity. It is of clinical interest.
Author Response
| headline 2.1 starts with a small letter.. micro-CT... | Done |
| formulas 2 + 3 really should have the angle in radians, one might think degrees is Ok. The authors sure know this. | It was chosen that to improve interpretation of this fitting that degrees would be used it would be much more intuitive to determine the change in degrees that radians. Also, this does not affect the mathematical operation in this case as in this formula it would be just an additional constant. |
Reviewer 3 Report
This is a very interesting article that describes the influence of scala tympani conformation on cochlear implantation insertion forces. The overall work is novel and interesting, some improvement in writing would be helpful - and a significant improvement in the clarity of the figures is absolutely required.
Specific comments:
Introduction
49 - Acoustic?
53 - Typically?
96 - nor, or rephrase the sentence.
Methods
104 - microCT?
109 - Points 'added' to the BM? Please explain this further.
Results
254 - incorrect labels
264 - Figure A2? The figure legends are confusing and inconsistent. Please improve the readability of all figures - figure legends, axis labels, text size, etc.
288 - Cochlear size is introduced here, but this is very important. An improvement should be made here to add cochlear size at the introduction.
343 - NP1 / NP2 introduced in text, but acronym not declared.
Discussion:
Discussion is good, general wording/grammar improvements needed.
Author Response
| Introduction | |
| 49 - Acoustic? | Corrected |
| 53 - Typically? | Corrected |
| 96 - nor, or rephrase the sentence. | Corrected |
| Methods | |
| 104 - microCT? | As both microCT and micro-CT are used, we decided to be consistent with the later form. |
| 109 - Points 'added' to the BM? Please explain this further. | Corrected |
| Results | |
| 254 - incorrect labels | |
| 264 - Figure A2? The figure legends are confusing and inconsistent. Please improve the readability of all figures - figure legends, axis labels, text size, etc. | Corrected |
| 288 - Cochlear size is introduced here, but this is very important. An improvement should be made here to add cochlear size at the introduction. | Cochlear size clarified in manuscript |
| 343 - NP1 / NP2 introduced in text, but acronym not declared. | Declared in line 154 and now restated here to improve comprehension |
| Discussion: | |
| Discussion is good, general wording/grammar improvements needed. | Edited discussion to improve readability |
Round 2
Reviewer 1 Report
The authors have responded to all the previous comments. The manuscript is corrected according to the suggestions. The figures, narrative and conclusions are concise and complete.